# Challenges and Trends in User Trust Discourse in AI Popularity

**Sonia Sousa** [1,2,3,*,†] [iD], **José Cravino** [2,4,†] [iD] and **Paulo Martins** [2,3] [iD]

1   School of Digital Technologies, Tallinn University, Narva Mnt 25, 10120 Tallinn, Estonia
2   Escola de Ciências e Tecnologia, Universidade de Trás-os-Montes e Alto Douro, 5000-801 Vila Real, Portugal
3   INESC TEC-Institute for Systems and Computer Engineering, Technology and Science,
    4000-008 Porto, Portugal
4   CIDTFF—Centro de Investigação em Didática e Tecnologia na Formação de Formadores,
    Universidade de Aveiro, 3810-193 Aveiro, Portugal
*   Correspondence: scs@tlu.ee
†   Writting All and conceptual S.S.

**Abstract:** The Internet revolution in 1990, followed by the data-driven and information revolution, has transformed the world as we know it. Nowadays, what seam to be 10 to 20 years ago, a science fiction idea (i.e., machines dominating the world) is seen as possible. This revolution also brought a need for new regulatory practices where user trust and artificial Intelligence (AI) discourse has a central role. This work aims to clarify some misconceptions about user trust in AI discourse and fight the tendency to design vulnerable interactions that lead to further breaches of trust, both real and perceived. Findings illustrate the lack of clarity in understanding user trust and its effects on computer science, especially in measuring user trust characteristics. It argues for clarifying those notions to avoid possible trust gaps and misinterpretations in AI adoption and appropriation.

**Keywords:** human-computer Interaction; trust; human-to-human relationship; human-to-technology relationship

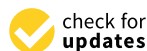



## 1. Introduction

Current digital transformation events forced new digital interaction patterns that did not exist 10 or 20 years ago, impacting how we play, work, and build communities. As a result, society, organizations, and individuals must rapidly adapt and adjust to these new digital norms and behaviours. In the short term, the division between online and physical activities diminished, increasing the capacity to act in a larger digital market and society. Consequently, these digital transformation events forced us to become more vulnerable to the actions of digital parties without adequately understanding or being able to assess their risk, competency, and intentions. Digital social media platforms like Facebook (2004), Twitter (2006), and Youtube (2005) or messaging services such as WhatsApp (2009) have promoted this new era of communication that resulted in continuous attempts to subvert their original purposes (i.e., malicious acts). Examples of this can be the generation of mistrust against vaccines, the creation of content supporting climate denial theories or disinformation campaigns, or the surge in data breaches [1–5].

This attempt of '*science and engineering of making intelligent machines*', as McCarthy [6] conceptualized Artificial intelligence (AI), resulted in the spread of software that uses computer technology to generate outputs such as content, predictions, recommendations, or decisions influencing the environment they interact. Moreover, adding to this notion of AI shared by the European Commission's Artificial Intelligence Act (https://artificialintelligenceact.com/, accessed on 6 December 2022) is the fact that nowadays, we can find AI systems that mimic, think and act like humans [7]. What highlights the potential of those AI mechanisms, not just as information revolution tools but as well as data threats, take the example reported in books like '*Weapons of math destruction: How big data increases inequality and threatens democracy*', or '*The AI delusion*'.

Therefore, what once was seen as science fiction has become a reality, emphasising people's fears and mistrust of the possibility of machines dominating the world. Those fears have led to the surge of new reports, articles, and principles that seek more trustworthy AI (TAI) visions that provide a Human-centered vision of users' trust and socio-ethical characteristics towards AI [5,8–12]. It also increased the discourse on AI and the need to complement existing data protection regulatory mechanisms (e.g., GDPR-ISO/IEC 27001) https://gdpr-info.eu/, accessed on 6 December 2022). It also highlights the need for seeking new 'responsible' AI ethical solutions that are less technical and profit-oriented.

This Human-centered vision of AI, however, is hard to understand, especially if, like Bryson [13], we try to make accountable the service providers, in fact, to the detriment of the default, mainstream attention is given to it for AI providers, developers, and designers, it is unclear how to ensure that the AI system designed by humans can be reliable, safe, and trustworthy [14]. More, AI's complexity makes it challenging to guarantee that AI is not prone to incorrect decisions or malevolent surveillance practices. Like the GDPR, as AI's popularity increase, it also increases its potential to create opaque practices and harmful effects and AI's unpredictability, making it hard to be audited, externally verified, or question (i.e., black box) [15]. Additionally, AI's unpredictability makes it difficult to avoid unforeseen digital transformations, harmful practices, and risks. It also makes it hard to predict behaviour and identify errors that can lead to biased decisions, unforeseen risks, and fears [10,16–19].

In conclusion, the increase in AI's popularity also increased its complexity, the number of decentralized and distributed systems solutions, increased as well the AI's opacity and unpredictability. When mixed with poor design practices, these AI characteristics can produce vulnerable interactions that lead to further breaches of trust (both real and perceived). With this work, we aim to share our vision regarding the challenges and trends in user trust discourse in AI popularity from a Human-Computer Interaction (HCI) perspective. Results presented are supported by the author's work in mapping the trust implications in HCI during the past decade and situated in the context of three recent systematic literature reviews performed on trust and technology [20–22]. Hoping that this clarifies the nature of user trust in recent AI discourse (RQ) and also avoids designing vulnerable AI artefacts that build on trust without understanding its influence in the uptake and appropriation of those AI complex systems. This work's main contribution is to link the previous trust in technology discourse with recent AI popularity and user trust trends. Then, it illustrates the importance of providing an HCI perspective of user trust discourse. Finally, establish a link between past trust in technology practices, current thoughts, and research gaps.

### 1.1. AI Popularity and the Discourse on Users' Trust

The recent waves of technology innovations are marked by AI popularity, the social network revolution, distributed data, automated decision-making, and the ability to trace and persuade behaviours [18,23–26]. These AI information-driven revolutions recently resulted in the spread of AI complex and distributed software solutions that generate automated and unpredictable outputs that cannot guarantee that they are not prone to provide incorrect content, predictions, or recommendations or mislead people into incorrect decisions that can have potentially harmful consequences in environments they interact, like malevolent surveillance practices and disinformation practices [18,27–32].

This technological revolution wave also resulted, in an increased discourse toward trust in AI, seeking solutions to regulate, diminish people's fears, and guarantee a user trust approach to the topic. Take the example of the European Commission draft EU AI act (https://artificialintelligenceact.com/, accessed on 6 December 2022), the Organisation for Economic Co-operation and Development (OECD), the Business Machines Corporation (IBM) and their efforts to clarify the Trustworthy AI (TAI) principles and practices [11,33,34].

This increase and new TAI discourse challenge HCI practitioners, a need to establish new trust boundaries (e.g., regulations, socio-ethical practices, etc.) to ensure Humans' abil-

ity to monitor or control their actions [16,35]. However, like AI, addressing trust in technology can be a complex subject for non-experts, as it acknowledges the deterministic models (that aggregate system technical characteristics) and the human-social characteristics that envision trust through a set of indirect parameters. This has raised another challenge to AI popularity, seeking solutions to trigger users' trust in AI [10,16–19]. However, with the increased popularity of AI software, it is unavoidable for society to be susceptible to its opaque practices, which can lead to further breaches of trust. Adding to this, the fast spread and dependency on AI prevent individuals from fully grasping the intricate complexity of those machines' capabilities, which can lead to potentially harmful consequences. Consequently, we believe that a new trend in user trust research will be revealed, similar to the rise of the Special Interest Group in 1982, to address the need for the design of Human-Computer Interactions (e.g., SIGCHI). This will lead to new international standards, expert groups, and international norms to tackle this problem. Take the example of the high-level expert group (AIHLEG) (https://digital-strategy.ec.europa.eu/en/policies/expert-group-ai, accessed on 6 December 2022) [34]. Or the Working group 3—trustworthiness referred to in the international standards for Artificial intelligence (ISO/IEC JTC 1/SC 42/WG 3) (https://www.iso.org/committee/6794475.html, accessed on 6 December 2022).

The above initiatives attempt to defy the surveillance capitalism practices and fight the corporate approach to data as the new oil of this century, seeking short-term profits without considering the ethical consequences of their actions. However, their focus is on tackling the AI problem and not so much focus on understanding or mapping the influence or consequences of user trust in its adoption and appropriation practices. The recent EU's AI act (https://artificialintelligenceact.com/, accessed on 6 December 2022) shares a broader vision of this problem and represents an attempt to incorporate the notions of risk and trust within AI's characteristics like complexity, opacity, unpredictability, autonomy, and data-driven qualities. highlighting the need for finding new user trust solutions that foster feelings of safety, control, and trustworthiness in current AI solutions.

In their regulatory scope (i.e., ethical guidelines for trustworthy AI), the EU encourage building public trust as an investment for ensuring AI innovation and respecting fundamental rights and European values. It also classifies trust as a need to be understood within four potential AI risks to health, safety, and fundamental rights from minimal or no risk, AI with specific transparency obligations (e.g., 'Impersonation' (bots)), High risk (e.g., recruitment, medical Devices), and an unacceptable risk (e.g., social scoring).

Those demand different Trustworthy AI (TAI) frames to ensure public trust in AI computing speech or facial recognition techniques in applications like social chatbots, human-robot interaction, etc. For AI providers and non-expert in trust, however, it is challenging to fully understand the user trust influence in AI acceptance and appropriation, as current, trustworthy AI principles provide a set of requirements and obligations with an unclear trust notion, sometimes associated with notions of ethics and accountability. In sum, for now, the EU's AI act is a very recent regulatory framework, but those principles are likely to be extended to the world, similarly to the GDPR. If so, it becomes unavoidable to clarify the nature of user trust in recent AI discourse (RQ1). Including clarifying the link between past trust in technology practices, current thoughts, and research gaps.

*1.2. TAI Conceptual Challenges*

The above-described misconceptions and malevolent practices, followed by the EU AI Act draft and adopting a risk-based approach (unacceptable risk, high risk, & limited or minimal risk), raised the need for addressing the challenges and trends in user trust discourse in AI. As well as for providing further conceptual clarifications and strategies that demystify the discourse of trust and socio-ethical considerations, user characteristics when facing risk-based decisions, and design and technical implementations of trust [22,36,37]. Avoiding trust in AI solutions that are marrow framed from technical or single constructs like explainable AI (XAI), privacy, security, or computational trust. That eventually cannot

guarantee that humans do not misinterpret the causality of complex systems with which they interact and lead to further breaches of trust and fears [27,38].

Take, for instance, the following socio-ethical considerations design toolkits, guidelines, checklists, and frameworks whose goal is to bring more clarity to the problem. Like the IDEO toolkits to established by the entitled trust Catalyst Fund (https://www.ideo.com/post/ai-ethics-collaborative-activities-for-designers, accessed on 6 December 2022), and the IBM Trustworthy AI, a human-centered approach (https://www.ibm.com/watson/trustworthy-ai, accessed on 6 December 2022), or [39], an agile framework for producing trustworthy AI was detailed in [40], and an article entitled Human-centered artificial intelligence: Reliable, safe & trustworthy was presented by Shneiderman (2020) [16]. Similarly the Assessment List for Trustworthy Artificial Intelligence (ALTAI) (https://altai.insight-centre.org/, accessed on 6 December 2022), and the EU Ethics guidelines for trustworthy AI (https://op.europa.eu/en/publication-detail/-/publication/d3988569-0434-11ea-8c1f-01aa75ed71a1, accessed on 6 December 2022). They neither offer clarity on trust notions nor explain how the proposed practices leverage user trust in AI.

As Bach et al. [22] and Ajenaghughrure et al. [20] findings confirm, measuring trust remains challenging for researchers. Currently, there is more than one way to define and measure trust. According to Bach et al. [22], Out of the 23 empirical studies explored, only seven explicitly define trust. At the same time, eight conceptualize it, and the remaining nine provide neither. Therefore, user trust is still an underexplored topic. According to Ajenaghughrure et al. [20], there is still a lack of clarity on measuring trust in real-time, and few solutions provide stable and accurate ensemble models (results from 51 publications). Those that exist are narrow, subjective, and context-specific, leading to an oversupply of models lowering the adoption. Especially when using psychophysiological signals to measure trust in real time.

This phenomenon happens despite computational trust research emerging in 2000 as a need to provide a technical-centred and automated way to tackle the trust factors in system design, i.e., to authenticate trustee's reputation and credibility [41]. Ultimately, and to avoid the past mistake of forward-push regulations, trust measures that ultimately are technically implemented without considering the tensions between the current state of creating new technical innovations, profit-oriented deployment, and its socio-technical implications across societies. Researchers need to look beyond the technical-centred vision of trust in computing and produce new user trust notions that help clarify the role of trust in these new technical profit-oriented AI innovations. As Rossi [42] (p. 132) argues, to fully gauge AI potential benefits, trust needs to be established, both in the technology itself and in those who produce it, and to put such principles to work, we need robust implementation mechanisms. Yet, researchers need to ensure its proper application by providing frameworks that clarify its implementation and avoid misinterpretation and misguided implementations [16,19]. Claiming once more for a shift from emphasis system's technical qualities toward human-centred qualities, similar to the move between usability and user experiences, i.e., from a focus on design features towards a focus on experiences of use [43–45].

## 2. Discussion

The challenges mentioned above have shifted current literature discourse towards Human-centered design (HCD) as a strategy to address the socio-technical characteristics of design features and lessen misinterpretation gaps in regulatory practices. These needs are followed by a need to clarify the current trust lenses of analysis, as trust can be a key to lessening the risks to the development, deployment, and use of AI in the EU or when it will affect people in the EU. However, as seen in the literature, trust divergent notions can prevent non-experts from adequately addressing this need from an HCD perspective, which can lead to an increased risk of failure, increasing its misinterpretation gaps that can be more harmful than good [46].

Therefore, needs and challenges that were not recognized 10 to 20 years ago are now a reality, which can create gaps in IT education. Currently, few curricula contemplate this socio-technical view or Human-centered design vision nor the ethical focus on measuring the risks of their potential misuse. As a result, IT and AI specialists might not be equipped with the necessary skills to address the challenges mentioned above, let alone know how to deal with this topic's complexity and application challenges, i.e., the Trustworthy AI (TAI) risk-based approach promoted by the EU. In that regard, despite agreeing that HCI researchers can contribute to broadening this analysis and helping IT, specialists adopt more user-centred strategies to prevent building systems that can lead to risky, unsafe life or long-term threatening social ramifications like the examples presented above [16,47]. They also need novel theories and validated strategies to address the socio-technical effects of trust in System complexity.

Like in the past, the focus shifted from measuring the usability characteristics of a system (e.g., efficiency and effectiveness) towards or focusing on hedonic characteristics (e.g., emotion and satisfaction), and now to a risk-based approach where trust is part of users' experiences. However, this needs to be followed by clear notions of trust, psychologically validated analysis, and associated underlying theories in context [36,42,43,48]. Trust, like satisfaction, is a human characteristic, not a machine characteristic. Past narrow views and assumptions on trust in technology might not fit in current Human-centered TAI applications [36]. A vision highlighted and shared in Figure 1, based on a culmination of various works performed in the past ten years (e.g., literature reviews, participatory research, teaching, supervising, etc.) to understand the nature of user trust in Human-Computer Interaction (HCI) [35,49–56].

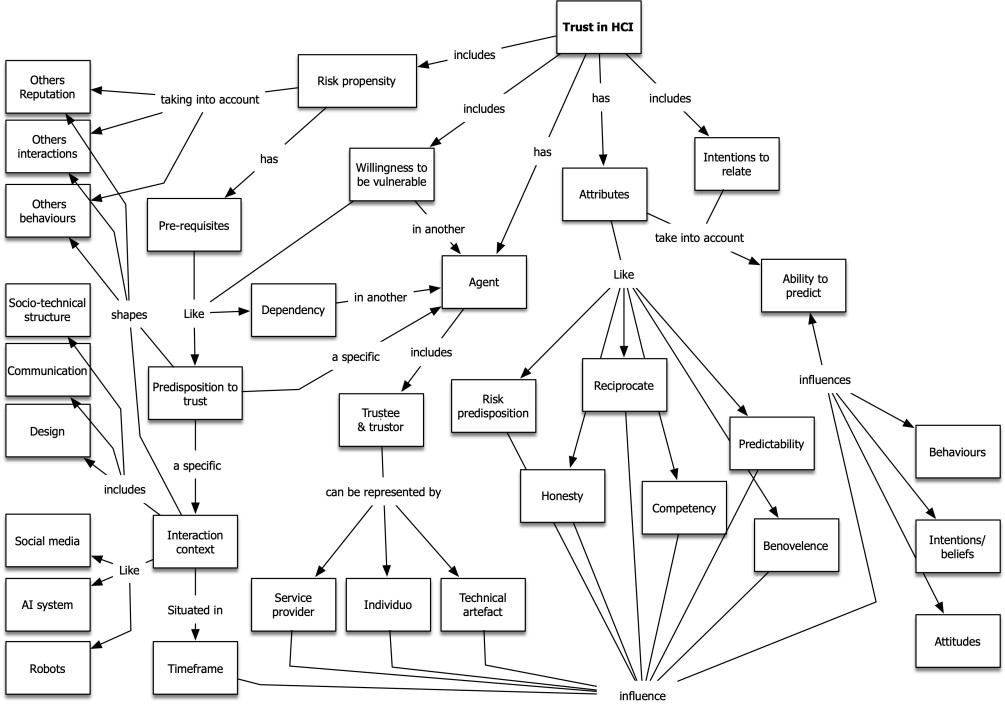

**Figure 1.** The nature of trust in HCI: Conceptualization.

*The Nature of Trust Research in HCI*

Trust in HCI, as illustrated in Figure 1, is a social-technical quality that supports the interrelationship between a trustee (the entity being trusted) and the trustor (the person who trusts). Trusting is a will to be vulnerable. Note that vulnerability implies risk acceptance based on how well the 'trustor' can perceive the 'trustee' as trustworthy, as Mayer et al. [35]. However, past views of trust tend to associate it with single constructs, like 'credibility', 'privacy', and 'security'. Associating user trust as a characteristic of disclosure of certain

types of information (i.e., privacy) or preventing cybersecurity threats to ensure system trustworthiness [57].

Maybe a reason why trust notions and applications in computer science literature provide an oversupply of trust visions, solutions, and applications. Take the example of Sousa et al. [21] findings (results from 69 publications) that reveal that trust notions and solutions can differ and depend on the application context. Mainly trust is addressed as a quality to influence technology adoption, data sharing credibility, and positively influencing user's intentions and behaviours. Take, for example, how trust is addressed within the privacy and security research topic. Herein, researchers see trust as avoiding potential misuse and access to personal data. It is sometimes mentioned as an information credibility attribute. Trust visions in e-Commerce, eHealth, or eGovernment are connected with 'risk', 'reputation', and 'assurance' mechanisms to establish loyalty, minimize risk and uncertainty and support decision-making. Solutions range from data-driven trust models to observing the impact of trust in encouraging decision-making and encouraging technology adoption (e.g., commercial transactions, diagnostic tools, adoption of services, etc.). In social networks, trust emerged as a way to sustain interaction processes between members of actor networks in emerging scenarios and argue that trust contributes to promoting the regulation of interaction processes. Trust is also useful in creating sustainable computer support collaborative work to support interpersonal interactions online.

Regarding its associated concepts, trust is associated with transparency, assurances, data credibility, technical and design feature, trustworthiness, users' predispositions to trust, explicability, etc. Mainly ways to reduce the uncertainty and risk of unpredictable and opaque systems, e.g., speech and facial recognition systems, crewless aerial vehicles (e.g., drones), IoT applications, or human-robot interactions (HRI). However, most trust studies present a narrow and simplified view, focusing on data-driven computational trust mechanisms to rate a system or a person as reliable. Presenting a view of trust as rational expectation, person, object, or good reliability or credibility when a first encounter occurs and no trust has been established, i.e., establish trust between two strangers. Discarding more complex aspects o trusted relations through time, the Human-system relationship is established through various indirect attributes like risk perceptions, competency, and benevolence [52,58–61].

The above paragraph illustrates the pertinence of providing new user trust visions that can be adjusted to new digital AI regulations, behaviours, and innovations. It also illustrates the complexity of both subjects, AI and user trust. On the one hand, the new EU AI act sees public trust as a way to guarantee AI innovations, guaranteeing that it is not prone to high risks like leading users to incorrect decisions or malevolent surveillance practices. On another, the AI providers are not experts in trust in technology, which make it hard for them to acknowledge the deterministic models (that aggregate system technical characteristics) and the human-social characteristics that envision trust through a set of indirect parameters. In literature, for instance, trust is associated with narrow views like 'reputation', 'privacy', and 'security'. Literature on trust and computing also comes associated with computational trust model [62].

With the same regard to security and privacy measures and their role in fostering AI trustworthiness, recent malevolent use demonstrates that new visions need to be adjusted to prevent mistrust in technology. Just addressing trustworthy AI measures as a way of preventing intrusion, allowing the individual the right to preserve the confidentiality, integrity, and availability of information might not be enough within today's socio-technical complexity [63,64]. Privacy refers to the individual's right to determine how, when, and to what extent they will disclose their data to another person or an organization [65]. As Figure 2 illustrates, user trust considers Socio-ethical considerations, Technical artefact, Application context, and Trustee & trustor characteristics [21,22,66–69].

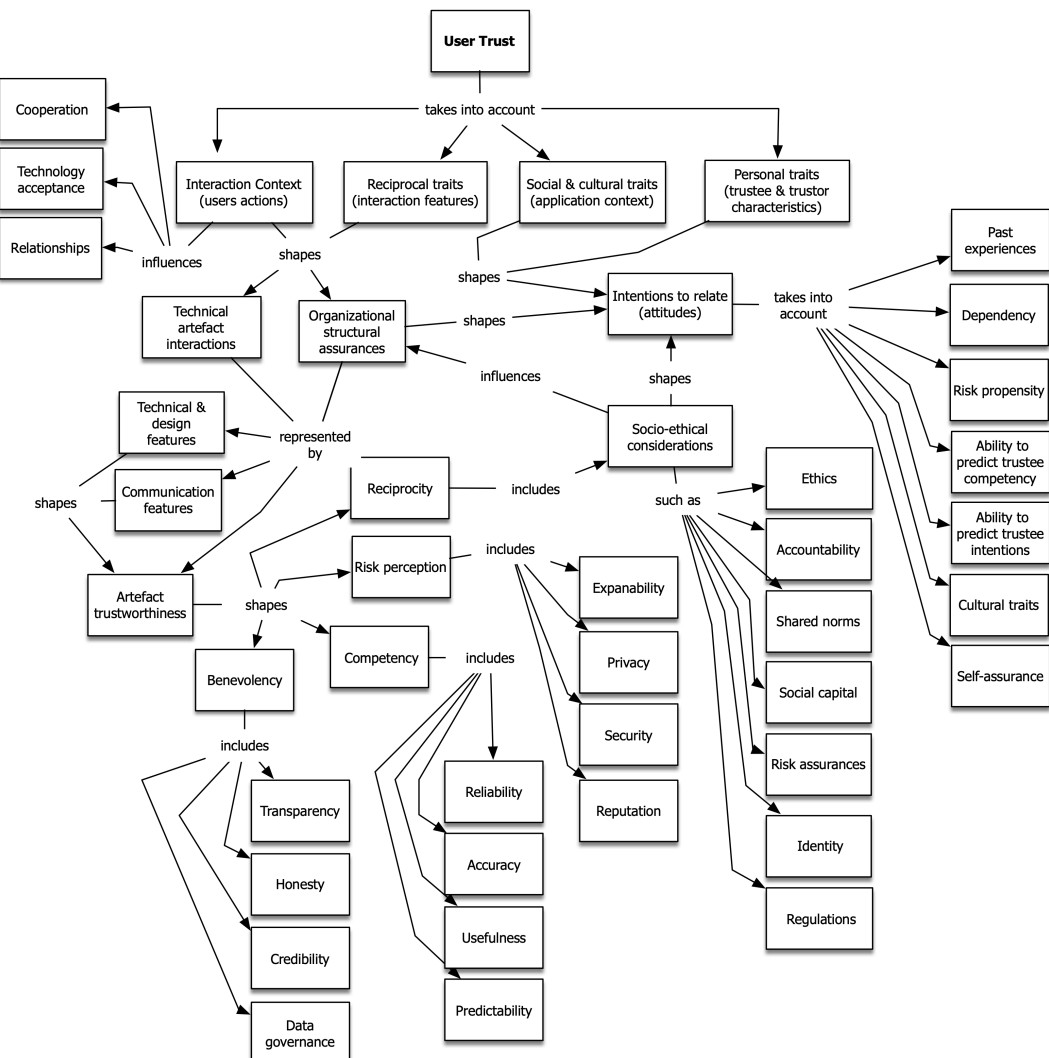

**Figure 2.** The user trust influence in shaping the interaction context: Conceptualization.

A trustful relationship requires assessing the risks (i.e., gains and losses). Requires evaluates the tool's ability (e.g., competence, reliability, accuracy, etc.) to perform the desired outcomes; and assesses if an entity (i.e., company, organization, institution, etc.). Requires individuals exceptions that digital relationships follow expected social norms and ethical principles. For instance, trustworthiness is an attribute or quality of a system, a person, or an organizational structure. As the Oxford dictionary describes it, **Trustworthiness** is the quality or fact of being trustworthy (=able to be trusted). Work like the Assessment List for Trustworthy Artificial Intelligence (ALTAI), or Trustworthy AI (TAI), is human-centred, or even Human-centered artificial intelligence: Reliable, safe & trustworthy do not address it or only address it from a shallow view.

On the other hand, if **Trust** is to believe that someone is good and honest and will not harm you or that something is safe and reliable. **Trustworthy**, on the other hand, is the ability to be trusted, and trustworthiness is a judgment of trust. Trusting reflects the strength of a person's belief that someone else will help them to achieve an acceptable outcome [70,71]. Trustworthiness and trustworthy are characteristics of trust, and in any complex construct, both (qualities and characteristics) are measured through indirect and direct interrelated factors TAI regulations are one example.

Measuring the attribute or quality of a system (e.g., privacy or security) might not be enough to address it. Or, for instance, take the example of system explainability (XAI) or computational trust models called by some reputation mechanisms [62,72]. As Davis et al. [27] claim, some technical-centred explanations might mislead individuals'

understanding of the subject. As Dörner [38] work illustrates, in some cases, humans' limitations to understanding a complex subject might prevent them from misunderstanding their work. As Sousa [73] result revealed, the interrelations between trust and performance can be negative, i.e., the higher the trust, the lower the performance. Yet, some limited-risk applications do not prevent them from using and benefiting from these tools. I do not need to understand a car's or aeroplane's mechanics to trust and use it. Individuals already (successfully) interact with complex AI systems daily. But I should be aware of its potential threats to making knowledgeable decisions, especially when adopting an AI system leads to an unacceptable or high-risk approach as the EU act describes it. Thus, to successfully maintain trust in specific events, we should not look at it from a narrow technical perspective. User trust in AI (i.e., technical artefact) can also be influenced by users' cultural traits, past experiences, and applications context [36,42].

Therefore, it is important to include both visions: **Trust as a personal trait**, understood as a perceived skill set or competencies of trustee characteristics (e.g., how teachers are perceived in a school system); **Trust as a social trait**, understood as the mutual "faithfulness" on which all social relationships ultimately depend. **Trust** reflects an attitude, or observable behaviour, i.e., the extent to which a trustee can be perceived in society. For instance, to want to do good, be honest – 'benevolence.' Follow privacy and security regulations. **Trust as reciprocal trait**, closely related with ethical and fair. For instance, the extent to which the trustee adheres to a set of principles that the trustor finds acceptable—for instance, an economic transaction.

This led to another application challenge, trust measurements [21,22]. Current trust misconceptions lead to an oversupply of computational trust assessment models that can only be used in narrow AI applications. Han et al. [74] recommendation trust model for P2P networks and Hoffman et al. [75] trust beyond security: an expanded trust model is an example of that. Some, however, measures of trust across gender stereotyping and self-esteem indicate that trust can be measured in a broader socio-technical perspective [76,77]. The EU self-assessment mechanisms for Trustworthy Artificial Intelligence created by the AIHLEG expert group is another example [78] of broadening the view. The same regards the Human-Computer trust (HTC) psychometric scales proposed by Madsen and Gregor [79], SHAPE Automation trust Index (SATI) [80]. We need new HCI mechanisms to measure potentially faulty TAI design practices, which can lead to risky, unsafe life or threatening social ramifications [16,47]. If not, HCI researchers might continue looking for specific and narrow solutions that can fail when applied in broader contexts. An example is the latest pursuit of AI system explainability (XAI) or computational trust models might not be enough to foster trust in users. On the contrary, some technical-centred explanations might mislead individuals' understanding of the subject. Or, some computational trust models are so narrow in their application that they might successfully maintain the initial trust formation in e-commerce but are not valuable for e-health.

Generally, looking at the above concepts, many researchers understand trust as a specific quality of the relationship between a trustee (the entity being trusted) and the trustor (the person who trusts). In other words, the trust dynamic contemplates a subtle decision based on the game's complexity that they find themselves playing, as Bachrach and Zizzo [81] and Luhmann [56] describe it. However above paragraph also represents the need for the trustor (i.e., human) to perceive the trustee as trustworthy. For instance, this system can have all the technical mechanisms to be secure, but if the trustor cannot see those who see these mechanisms in action, they might perceive that they are not in place. So, trustworthiness and being trustworthy are two complementary aspects of trust. Perceived trustworthiness is an individual's assessment of how much an object, a person, or an organization, can be trusted. People assess trustworthiness based on information they perceive and or receive influenced by cultural and past experiences, and both qualities can evolve through time. In conclusion, in the Socio-ethical AI context, trust notions still need further clarification to ensure that solutions foster public trust and fundamental rights for minimal or no risk in the AI data protection process and non-bias

(see EU's AI act). Same regards how we connect trust with information credibility and ethical practices. As well as studying the socio-ethical AI implications (i.e., explainable AI) in the acceptance and use of the technology. Or even when using a trust to seek more control in automated and intelligent system predictions. Or, provide socio-ethical AI as transparency and responsibility solutions through trusted agencies and other audition mechanisms [21,82].

## 3. Conclusions

The first digital revolution, i.e., Internet revolution in 1990, has brought big changes to how we communicate and interact across-country. However, recent digital revolutions characterised by the data-driven and information revolutions transformed the world and society as we know it. AI systems enabled by the social network, followed by the ability to trace and persuade behaviours, have altered social democratic practices and applications. The challenge nowadays is finding ways to adjust current regulatory practices to these new digital practices. Including looking for ways to fight the advancement of potential AI malpractices and minimize the risk of malevolent use.

The above findings reveal the importance of clarifying the user trust notions in recent AI discourse. This is to lessen possible misinterpretation of trust and notion gaps in these new ethical guidelines for trustworthy AI regulatory practices. This is to avoid misconceptions about user trust in AI discourse and fight the tendency to design vulnerable interactions that lead to further breaches of trust, both real and perceived. Provide also evidence of the lack of trust and understanding of computer science, especially in assessing trust user characteristics and user-centred perspectives Ajenaghughrure et al. [20], Sousa et al. [21], Bach et al. [22]. Also, frame the term 'trustworthy computing' as critical for technology adoption and complex system appropriations. As Shneiderman [16] stresses, we need to conceive a more Human-Centered Artificial Intelligence (HCAI) paradigm where human-machine relationships are perceived as safe, reliable, and trustworthy.

We are now acknowledging that despite the attention given to technical characteristics like 'privacy', 'security', or 'computational trust and reputation', malevolent technological practices still prevail. Also, widespread AI-driven applications and their associated risks bring new challenges to distrust and fear across-country discourse. Take the examples of persuasive malevolent design, deceptive designs, unethical business decisions associated with increasing concerns on socio-ethical AI, technical and design features, and user characteristics that Bach et al. [22] work to mention. Another challenge addressed is the human-likeness that misguides users to misplace the attributes of a trusted human, human-to-human exchanges mediated through technology and their trust in a human-artefact relationship [83–86]. Even though some researchers claim that 'people trust people, not technology, as technology does not possess moral agency and the ability to do right or wrong [56,87–90]. Researchers fail to acknowledge trust complexity and how its indirect measures affect users' trust perceptions in the system's adoption and appropriation.

After two decades of investment and advances, we now change the discourse towards a human-centred view and the need to develop, deploy and measure the quality of trust perceptions from an HCD perspective. Addressing AI-related attributes like reliability, safety, security, privacy, availability, usability, accuracy, robustness, fairness, accountability, transparency, interpretability, explainability, ethics, and trustworthiness.

**Funding:** This work was supported by the Trust and Influence Programme [FA8655-22-1-7051], the European Office of Aerospace Research and Development, and the US Air Force Office of Scientific Research. This study was partly funded by AI-Mind, which has received funding from the European Union's Horizon 2020 research and innovation programme under grant agreement No 964220.

**Data Availability Statement:** Not applicable.

**Conflicts of Interest:** The authors declare no conflict of interest.

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
