# Peer review of "Challenges and Trends in User Trust Discourse in AI Popularity"

_mti, doi:10.3390/mti7020013_

Round 1
Reviewer 1 Report
In the paper, interesting research outcomes are presented. The Authors have provided an overview of the nature of trust research in Human-Computer Interaction and have discussed trust-related issues like practices, current research rationale, and challenges. In my opinion, the undertaken research is important and needed nowadays. The paper is technically sound, well organized, and written on a good level. I appreciate the effort put into preparing Figure 1. I think it shows the concept of trust and the relationship between its artifacts in a legible and very interesting way.
However, the paper needs to be improved.
1) The authors’ contribution should be presented more clearly (e.g., in the Introduction section). In my opinion the outcomes presented in section 1.2 (Trust applications in computer science) are very similar to the section 3.1 in “Sousa, S.; Cravino, J.; Lamas, D.; Martins, P. Confiança e tecnologia: práticas, conceitos e ferramentas. Revista Ibérica de Sistemas e Tecnologias de Informação 2021, 45, 146–164” [59]. Therefore, I have great difficulty finding novelty elements and Authors’ contributions in this paper.
2) Second, some minor mistakes should be corrected:
a) First section (Introduction) should start with the number 1 not 0.
b) In line 26 there is “way o life” and should be “way of life”.
c) In line 136 there is a lack of references (“[?]”).
d) In line 137 the text is too long (the text goes to the margin).
e) Sometimes there is a lack of space before references, like in lines 137, 141, 195,
f) In line 295 there is “represent” and should be “represents”.
g) In line 304 there is mentioned about ten major areas, but I see only 8 areas listed.
h) In line 321 there is “ensure’s” and probably should be “ensures”.
i) In line 362 there is a lack of a comma after “e.g.”.
j) In line 380 there is “mention” and should be “mentions”.
k) In line 425 there is “[80][68,81–83]” and should be rather “[68,80–83]”.
l) In the paper sometimes British English and sometimes American English is used (there is a difference in the spelling of words like “behavior”, “analyzed”, “summarize”, “organization” etc.) – please unify it.
Author Response
Dear Reviewer 1,
We would like to thank you for the valuable comments to improve our manuscript titled “Challenges and Trends in User Trust discourse in AI popularity ” for consideration in the special issue of Multimodal Technologies and Interaction (ISSN 2414- 4088). SPECIAL ISSUE "Multimodal User Interfaces and Experiences: Challenges, Applications, and Perspectives".
We have addressed all points we are confident that our manuscript has improved significantly to allow your readers to learn more about user trust in the AI discourse.
Unfortunately, I was not able to understand how the “Track Changes” worked in LaTeX, so we added the changes as a comment in the text % ***** CHANGES:
Thank you for your consideration of the manuscript.
Sincerely,

Reviewer 2 Report
This is an interesting paper with some potential. However there are some huge jumps in logic that makes assumptions about reader knowledge in some areas. There is also no clear overarching research question being articulated at the start of the paper.
AI itself needs to be better introduced in the introduction section. Currently, the introduction is well written narratively with a discussion of the recent trends and impacts of big data, with little discussion of AI until the section on AI itself. However there should be a clearly articulated link between big data and AI, and also a clear definition of AI which is missing. In fact AI itself is not fully explained other than some case examples in the AI section, and making AI synonymous with automation. This is a major shortcoming. The article also does not itself explicity define Artificial Intelligence (AI).
Line 117 - 119: There is a sweeping statement that an assumption should be adopted with weak justification and generalisability. This needs to be discussed in depth and is somewhat problematic, as there are IT and AI specialists that are ethicist and recognise the need for legal and regulatory compliance. In fact, the following section on TAI belies the attempt of many practitioners, academics and policy makers are aware of such issues. While it is true that they still may not have addressed the trust issue, it does not mean that they are not 'ethical, lawful or human-centred design experts'.
There needs to be a clear methodology section that articulates how the conceptualisations and literature review was done in a paper such as this.
Figure 1 is central to this paper. However, it is unclear to the reader how this figure was derived, what it is based on and how the links are made, or if it has been adapted from current literature. Why is it presented in this way and where are the justifications of the links. It's impossibly to clearly verify the links between concepts in the current form and the Discussion section does not actually clarify these links and conceptualisations. For example, some of the concepts seem to almost be looping into each other without explanation. In another example, the 'influence' section has lines with no arrows; why is that? I believe that a key challenge here is that there isn't an attempt to map out the process of how trust is built as suggested in the earlier section. Currently, Figure 1 is a presented as a web of concepts rather than making sense of the development of trust which would help the paper.
In addition, there are references to RQ1 and SRQ1 only in the Conclusions, which assumes readers know what these are, without articulation in the paper itself.
Minor Typos
There are various typos that need closer attention. Some (but not all) examples include:
Line 26: "...we question even less this way o life..."
Line 36: "...All this Raised a need..."
Line 250: "...applications don't prevent..."
Line 266: "More information is provided in Chapter ??..."
Author Response
Dear Reviewer 2,
We would like to thank you for the valuable comments to improve our manuscript titled “Challenges and Trends in User Trust discourse in AI popularity ” for consideration in the special issue of Multimodal Technologies and Interaction (ISSN 2414- 4088). SPECIAL ISSUE "Multimodal User Interfaces and Experiences: Challenges, Applications, and Perspectives".
We have addressed all points we are confident that our manuscript has improved significantly to allow your readers to learn more about user trust in the AI discourse.
Unfortunately, I was not able to understand how the “Track Changes” worked in LaTeX, so we added the changes as a comment in the text % ***** CHANGES:
Thank you for your consideration of the manuscript.
Sincerely,

Round 2
Reviewer 1 Report
In the paper, interesting research outcomes are presented. The Authors have provided an overview of the nature of trust research in Human-Computer Interaction and have discussed trust-related issues like practices, current research rationale, and challenges. In my opinion, the undertaken research is important and needed nowadays. The paper is technically sound, well-organized, and written at a high level. It describes in a comprehensive way the aim and contributions. I appreciate the effort put into preparing the revised version of the manuscript. All my comments and suggestions were taken into account by the Authors in the revised version of the paper. Good job!
Author Response
Dear Reviewer,
We would like to thank you for your valuable comments and consideration. We greatly appreciate it. We confirm that we checked and corrected the manuscript's various typos.
Thank you for considering the manuscript.
Reviewer 2 Report
Thanks for the reviews and improvements.
Figure 2 is better now but I am still trying to work out the term 'like' and other terms that do not match between the discussion and the figure. It is really hard for a reader to follow because dis-similar terms are used in the figure and the discussion.
For example trust as the different traits as discussed in-text is hard to match to sections of Figure 2. Is Trust as personal trait related to trustee and trustor personal traits? Therefore, Figure 2 is still difficult to follow.
Various typos need to be checked including the following examples:
Line 303 Davis et al repeated
Line 309 'don't' should be 'do not'?
Line 386 Typo (repeat of Schneiderman)
Author Response
Dear Reviewer,
We would like to thank you for the valuable comments and considerations for improvement and say that we did our best to have addressed all your points.
We can confirm that we checked and corrected the manuscript's various typos. In addition, we decided to revise and improve Figure 2 to minimise the mismatch with text discussions, and we believe that doing that improved the manuscript significantly.
Thank you for considering the manuscript